# Bacteriolytic Potential of *Enterococcus* Phage iF6 Isolated from “Sextaphag^®^” Therapeutic Phage Cocktail and Properties of Its Endolysins, Gp82 and Gp84

**DOI:** 10.3390/v15030767

**Published:** 2023-03-16

**Authors:** Rustam M. Buzikov, Olesya A. Kazantseva, Emma G. Piligrimova, Natalya A. Ryabova, Andrey M. Shadrin

**Affiliations:** 1Laboratory of Bacteriophage Biology, G. K. Skryabin Institute of Biochemistry and Physiology of Microorganisms, Pushchino Scientific Center for Biological Research of the Russian Academy of Sciences, Federal Research Center, 142290 Pushchino, Russia; 2Institute of Protein Research RAS, 142290 Pushchino, Russia

**Keywords:** iF6, bacteriophage, *Shiekvirus*, *Kochikohdavirus*, endolysins, antibiotic resistance, phage therapy, *Enterococcus faecium*, *Enterococcus faecalis*

## Abstract

The number of infections caused by antibiotic-resistant strains of bacteria is growing by the year. The pathogenic bacterial species *Enterococcus faecalis* and *Enterococcus faecium* are among the high priority candidate targets for the development of new therapeutic antibacterial agents. One of the most promising antibacterial agents are bacteriophages. According to the WHO, two phage-based therapeutic cocktails and two medical drugs based on phage endolysins are currently undergoing clinical trials. In this paper, we describe the virulent bacteriophage iF6 and the properties of two of its endolysins. The chromosome of the iF6 phage is 156,592 bp long and contains two direct terminal repeats, each 2108 bp long. Phylogenetically, iF6 belongs to the *Schiekvirus* genus, whose representatives are described as phages with a high therapeutic potential. The phage demonstrated a high adsorption rate; about 90% of iF6 virions attached to the host cells within one minute after the phage was added. Two iF6 endolysins were able to lyse enterococci cultures in both logarithmic and stationary growth phases. Especially promising is the HU-Gp84 endolysin; it was active against 77% of enterococci strains tested and remained active even after 1 h incubation at 60 °C. Thus, iF6-like enterococci phages appear to be a promising platform for the selection and development of new candidates for phage therapy.

## 1. Introduction

In recent decades, the number of bacterial infections caused by antibiotic-resistant bacteria has been growing. It was estimated in 2019 that at least 700,000 people die each year of drug-resistant diseases [1]. Enterococci are one of these antibiotic-resistant bacteria, a member of the ESKAPE group identified by World Healthcare Organization (WHO) as a high priority pathogen for the development of new antibacterial agents [2].

Pathogenic strains of enterococci can cause endocarditis, wound fever, neonatal sepsis, meningitis and urinary tract infections [3,4,5]. In many recent works describing the range of bacterial pathogens associated with COVID-19 pneumonia, enterococci occupy one of the leading positions [6,7,8,9].

The decline in the discovery and development of new antibiotics forces us to look for alternative approaches, one of which is the use of bacteriophages. Bacteriophages are natural predators of bacteria. It is believed that every 24 h, phages cause the lysis of up to 20% of the entire bacterial diversity on Earth [10]. In the 2020 WHO overview of antibacterial agents in clinical and preclinical development [2], the use of two bacteriophage cocktails was reported, both against *E. coli* and *K. pneumoniae*. Those were the oral cocktail developed by Adaptive Phage Therapeutics and US Department of Defense in phase 1/2 of clinical trials and the intravenous cocktail developed by Locus Bioscience in a phase 1b trial [2].

Another approach to curing antibiotic-resistant bacterial infections is the use of recombinant bacteriolytic enzymes of bacteriophages, also known as endolysins [11]. In the 2020 WHO report, two endolysins were mentioned: “tonabacase”, developed by iNtRON Biotechnology and Roivant Sciences in phase 2a trials; and “exebacase”, developed by Contrafect in the phase 3 of clinical trials [2].

The scientific production company “Microgen” produces 14 different phage cocktails at the amount of two million doses per year (https://www.microgen.ru/en/products/bakteriofagi/ accessed on 13 March 2023). One of their phage cocktails that is already being used in medical practice and freely available in a number of countries is the “Sextaphag^®^” polyvalent pyobacteriophage (https://www.microgen.ru/en/products/bakteriofagi/sekstafag-piobakteriofag-polivalentnyy/ accessed on 13 March 2023). The cocktail has the ability to specifically lyse staphylococci, streptococci (including enterococci), *Proteus*, *Klebsiella pneumoniae*, *Pseudomonas aeruginosa* and *Escherichia coli*. In this work, we describe a novel enterococci-infecting bacteriophage, iF6, which was isolated from the “Sextaphag^®^” cocktail, as well as the properties of two endolysins of this phage.

## 2. Materials and Methods

### 2.1. Bacterial Strains and Growth Conditions

The bacterial strains used in this study were obtained from the All-Russian Collection of Microorganisms (VKM) (http://www.vkm.ru accessed on 13 March 2023) and from the All-Russian Collection of Industrial Microorganisms (VKPM) (http://www.vkm.ru/collecti.htm accessed on 13 March 2023). Lysogeny broth (LB: 10 g/L tryptone, 5 g/L yest extract, 10 g/L NaCl) and LB agar (1.5% *w*/*v* for bottom layer and 0.45% *w*/*v* for top layer) were used for bacterial and phage cultivation. The following reagents were used to prepare the lysogenic broth and the LB agar: tryptone (Dia-M, Moscow, Russia), yeast extract (Dia-M, Moscow, Russia), agar-agar (PanReac Applichem, Darmstadt, Germany), NaCl (PanReac Applichem, Darmstadt, Germany). *E. coli* and enterococci cultures were grown at 37 °C and 35 °C, respectively. The optical density (OD) of cell suspensions was measured at 590 nm using an Expert-003 photometer (Econics-Expert, Moscow, Russia).

### 2.2. Phage Isolation, Purification and Propagation

Phage iF6 was isolated from the commercial phage cocktail “Sextaphag^®^” (Microgen, https://www.microgen.ru/en/ accessed on 13 March 2023) (lot number П813, production date December 2016). For phage isolation, purification and propagation, we used the double-agar-overlay plaque assay [12] with some modifications. *E. faecium* FS86 [13], which is sensitive to the phage, was used as the host strain for propagation. Briefly, 10 µL of the “Sextaphag^®^” was mixed with 100 µL of FS86 culture in the early log phase (OD590 of 0.4) and molten LB agar to a final agar concentration of 0.45%. After gentle short-term vortexing, the mixture was poured onto the bottom agar layer (1.5% LB agar) and incubated at 35 °C overnight.

Subsequently, a single plaque was transferred into a test tube and gently rotated for 4 h with 1 mL of SM buffer (100 mM NaCl, 1 mM MgSO_4_, 50 mM Tris-HCl pH 7.5), 10 mM CaCl_2_, 10 mM MgSO_4_ and 10 μL chloroform. The extract was centrifuged for 10 min at 7000× *g*. The supernatant was mixed with 500 µL of LB, 5 µL of a FS86 culture (OD590 of 0.4), CaCl_2_ and MgSO_4_ were each added to final concentration of 10 mM, and the solution was incubated at 35 °C with shaking overnight. The resulting suspension was serially diluted and plated by the double-layer agar method. The plates were cultured overnight at 37 °C. Three consecutive passages were carried out.

In order to obtain a high-titer phage preparation, 100 µL of the phage suspension and 2 mL of an overnight FS86 culture were added to 50 mL of LB and incubated at 35 °C with shaking and periodic monitoring of the optical density. After lysis (when a significant decrease in the optical density was detected), 2.88 g of NaCl and 500 µL of chloroform were added, and the incubation was continued for 30 min. The cell debris was removed by centrifugation at 16,000× *g* for 10 min. For phage precipitation, the supernatant was mixed with polyethylene glycol (PEG) 8000 (PanReac Applichem, Darmstadt, Germany) to a final concentration of 10% and incubated at 4 °C for 1 h, followed by centrifugation at 13,000× *g* for 20 min. The resultant precipitate was resuspended in 5 mL of SM+ buffer (100 mM NaCl, 1 mM MgSO_4_, 50 mM Tris-HCl pH 7.5, 0.01% gelatin), filtered through a 0.22 µm sterile filter and stored at 4 °C.

The high-titer phage preparation was used to prepare a purified phage suspension by CsCl (Reachim, Russia) density-gradient centrifugation. Three mL of the phage preparation was overlayed on the preformed CsCl gradient (1.25 g/mL, 1.4 g/mL, 1.5 g/mL and 1.7 g/mL; 2.5 mL each). Centrifugation was carried out at 10 °C, 110,862.3× *g*, for 2.5 h in an L7-55 Beckman Coulter ultracentrifuge, using a SW 40 Ti rotor. The phage fraction was collected and dialyzed against SM buffer. The phage titer was 10^9^ plaque-forming units (PFU)/mL.

### 2.3. Transmission Electron Microscopy

Ten microliters of the CsCl-purified phage suspension (10^9^ PFU/mL) were applied to 400-mesh carbon-coated copper grids, negatively stained with 1% uranyl acetate and analyzed using a JEM-100C (JEOL, Akishima, Japan) transmission electron microscope at 80 kV accelerating voltage. Images were taken on Kodak film SO-163 (Kodak, Cat. \# 74144, Hatfield, PA, USA). Phage particle dimensions were measured using ImageJ version 1.53e [https://imagej.nih.gov/ij/index.html, accessed on 1 June 2021] in relation to the scale bar generated from the microscope [14].

### 2.4. Host Range

A host range analysis of the phage was performed using 26 *Enterococcus* strains and 3 strains from other genera of bacteria (Appendix A), as described previously [15] but with a difference in incubation conditions (for 24 h at 35 °C).

### 2.5. Thermal and pH Stability of Bacteriophage

To assess phage stability at various temperatures, aliquots of phage suspensions at a titer of 5 × 10^7^ PFU/mL were incubated in temperatures ranging from 5 to 95 °C for 1 h. Then, the phage suspensions were cooled at 4 °C for 10 min, diluted and enumerated using the double-layer agar method. The pH stability of the phage was determined using buffers with different pH values: 50 mM glycine–HCl buffer (pH values 2.2 and 3), 200 mM sodium acetate buffer (pH 4 and 5), 100 mM sodium phosphate buffer (pH 6, 7 and 8) and 50 mM glycine–NaOH buffer (pH 9 and 10). Aliquots of the phage suspensions were added to each solution to a final concentration of 5 × 10^7^ PFU/mL and incubated at 25 °C for 1 h, followed by diluting and quantifying the number of surviving phages by the double-layer agar method. The experiments were carried out as five independent trials. The results were visualized in GraphPad Prism 8.4.3 [16,17] as a box-and-whisker diagram, with a 5–95% confidence interval.

### 2.6. Killing Assay and the Effect of Ca^2+^ and Mg^2+^

The assay was conducted as described previously [18], except that *E. faecium* FS86 was grown at 35 °C for 6 h. The initial concentration of FS86 cultures was 1 × 10^8^ colony forming units per mL (CFU/mL), and the iF6 phage was added to the cultures at multiplicity of infection (MOI) values of 0.1, 1, 3, 5 and 10. To evaluate the effects of Ca^2+^ and Mg^2+^ on iF6 lytic activity, the phage was added to FS86 cultures grown in LB with additional 10 mM CaCl_2_ and/or 10 mM MgCl_2_. Non-infected *E. faecium* FS86 cultures were used as control samples. Three independent trials of the experiment were carried out. The resultant growth curves were visualized in GraphPad Prism 8.4.3 [16,17].

### 2.7. Phage Adsorption Assay

The adsorption assay was performed according to the protocol developed by Kropinski [12] as previously described [18], with some modifications. The initial concentrations of *E. faecium* FS86 and the iF6 phage were 5 × 10^6^ CFU/mL and 5 × 10^5^ PFU/mL, respectively. Incubation was performed at 35 °C. Aliquots were taken every minute. The results were presented as a fraction of the initial phage number and visualized in GraphPad Prism 8.4.3 with error bars representing the standard deviation from three trials [16,17].

### 2.8. One-Step Growth Curve

The one-step growth curve experiment was performed according to the protocol developed by Hyman [19] as previously described [18], with the difference that the initial concentrations of *E. faecium* FS86 and the iF6 phage were 5 × 10^6^ CFU/mL and 5 × 10^5^ PFU/mL, respectively. The adsorption step was set to be 1 min, based on the results of the “Phage adsorption assay” experiment. Incubation temperature was 35 °C. Aliquots were taken at 15 min intervals for 2 h. The growth curve was visualized in GraphPad Prism 8.4.3 [16,17], with error bars representing the standard deviation from three trials.

### 2.9. Genome Sequencing, Assembly and Sequence Analysis

After CsCl-purification and dialysis against SM buffer, the resultant iF6 phage preparation was treated with DNAse I (NEB, #M0303S, Ipswich, MA, USA) and RNAse A (Thermo Scientific, EN0531) according to the manufacturer’s instructions in order to remove free DNA and RNA. Phage DNA was then extracted using the standard phenol–chloroform extraction protocol described by Sambrook et al. [20]. The iF6 paired-end Illumina sequencing library was constructed using the KAPA HyperPlus Kits (Kapa Biosystems, Wilmington, MA, USA) and was sequenced on the Illumina MiSeq platform. The reads (SRA: SRR12487495) were trimmed and quality-filtered with BBduk, BBMap [21] and then quality-controlled with FastQC version 0.11.7 [22]. The genome assembly was performed with the SPAdes v.3.14.1 software [23] and resulted in a single contig with the average coverage of 751×. Open reading frames (ORFs) identified using RASTtk v.2.0 [24]. This was followed by functional annotation using an in-house Python script, enabling each protein product to be analyzed with BLASTp (NCBI) [25] and HHpred [26]. ARAGORN v1.2.41 [27] was used for tRNA and tmRNA gene detection. Circular genome visualizations were created using the BRIG software v.0.95 [28].

### 2.10. Comparative Genomics

In order to find related phages, a BLASTn search was performed using the whole iF6 genome sequence as the query. A linear comparison diagram showing the genomic identity between iF6 and the most closely related phages was created with EasyFig v.2.2.2. [29]. The number of shared proteins was computed using the GET_HOMOLOGUES software v.3.3.3 [30] with the COGtriangles algorithm [31] (-t 0 –C 75 -e). VICTOR was used for phylogenetic inference from the whole proteome sequences of iF6 and the closest viruses using the Genome-BLAST Distance Phylogeny method with the settings recommended for prokaryotic viruses [32]. The tree was rooted at the midpoint and visualized with FigTree v.1.4.4 [33].

### 2.11. Determination of DNA Packaging Strategy

The phage genome termini were identified in a standard restriction analysis [34], followed by rapid amplification of genomic ends (RAGE) [35] as described previously [18], with slight modifications. For the RAGE analysis, phage DNA was digested with the restriction enzymes EcoRV and XhoI. DNA fragments containing the genome termini were extracted from the electrophoresis gel using Cleanup mini (Evrogen, cat. no. BC023, Moscow, Russia). The purified fragments were used in a typical DNA-tailing reaction with terminal transferase (New England Biolab, Cat. # M0315L, Ipswich, MA, USA) following the protocol provided by the enzyme manufacturer. Next, two PCRs were carried out sequentially using the TaqSE DNA polymerase (SibEnzyme, Cat. # E314, Novosibirsk, Russia) and the pairs of oligonucleotides designed for the right and left ends of the iF6 genome (Appendix A). The 3′-end tailing fragments were used as DNA templates in the first PCR, and the product of the first PCR was used as the template in the second reaction. The PCR products of the second reaction were extracted from the electrophoresis gel and used for Sanger sequencing with the primers RACE-R-iF6_fwd3 5′- GGACAACGTACAGACAATTATACCG -3′ and RACE-L-iF6_rev2 5′- TTCTTTGTGTGGTCGTGCTC -3′ for the right and left ends of the genome, respectively.

### 2.12. Cloning of Endolysin Genes and Enzyme Purification

The *iF6_82* and *iF6_84* genes were PCR-amplified using the Q5 high-fidelity DNA polymerase (New England Biolabs, Cat.# M0491, Ipswich, MA, USA) (Appendix A). The PCR products were cloned into two expression vectors: pET33b (+), containing a C-terminal His6 tag (-H); and pHUE, containing an N-terminal His6 tag and ubiquitin (HU-). For pET33b (+)-constructions: The PCR products of *iF6_82* and *iF6_84* were digested with NcoI and NotI and inserted into the pET33b (+) vector that had been digested with the same enzymes. The resultant plasmids, pET33-gp82H and pET33-gp84H, were used to produce proteins Gp82-H and Gp84-H with the C-terminal amino acid sequence extension “-AAALEHHHHHH”.

For pHUE-constructions: The PCR products were digested with EcoRI and KpnI and inserted into the pHUE vector that had been digested with the same enzymes. The constructed plasmids, pHUE-HUgp82 and pHUE-HUgp84, were used to produce proteins HU-Gp82 and HU-Gp84 with the N-terminal amino acid sequence extension “MGSSHHHHHHSSGLVPRGSHMQIFVKTLTGKTITLEVEPSDTIENVKAKIQDKEGIPPDQQRLIFAGKQLEDGRTLSDYNIQKESTLHLVLRLRGGSEF-”.

The constructed plasmids were transformed into *E. coli* BL21 Star (DE3) for target protein expression. The seeding bacterial cells were diluted (1:100) in 100 mL of LB broth and grown to OD590 = 0.4 at 37 °C with shaking. The expression of target proteins was induced with 0.5 mM isopropyl β-D-1-thiogalactopyranoside (Sigma-Aldrich, St. Lusis, MD, USA), and incubation was continued for 6 h at 37 °C. The bacterial cells were pelleted by centrifugation at 8000× *g* and 6 °C for 10 min.

The precipitated bacterial cells were resuspended in 40 mL of ice-cold Buffer A (10 mM Tris-HCl pH 7.5, 0.5 M NaCl, 5% glycerol). The cell suspension was placed in an ice bath and sonicated three times at power intensity 44.23 W/cm^2^ for 30 s, with five-minute intervals for cooling, using a Q700 Sonicator (Qsonica LCC, Newtown, CT, USA). The obtained lysate was centrifuged at 11,500× *g*, 6 °C for 45 min and filtered through a 0.45-μm Millipore syringe filter. The filtered lysate was loaded onto a 1 mL Ni-chelating column (GE Healthcare, Upsala, Sweden). The column was washed with buffer A and buffer B5 (10 mM Tris-HCl pH 7.5, 0.5 M NaCl, 50 mM imidazole, 5% glycerol). The proteins were eluted with 1 mL of imidazole elution buffer B10 (10 mM Tris-HCl pH 7.5, 0.5 M NaCl, 100 mM imidazole, 5% glycerol). Purification quality was evaluated by SDS polyacrylamide gel electrophoresis (SDS-PAGE). Chromatographic fractions containing the target proteins were concentrated and transferred into storage buffer D via three rounds of dialysis (10 mM Tris-HCl (pH 7.5), 0.5 M NaCl, 5% glycerol) at 4 °C for 12 h. Protein concentrations were determined by measuring adsorption at 280 nm (extinction coefficient = 1) using NanoPhotometer Pearl P-360 (Implen GmbH, Munich, Germany). Protein purity was assessed by SDS-PAGE. The final concentration of the purified enzymes was adjusted to 300 μg/mL, and dithiothreitol (DTT; Sigma-Aldrich, St. Lusis, MD, USA) was added to the enzymes to a final concentration of 10 mM. The enzymes were stored in 50 µL aliquots at −80 °C. For all subsequent experiments, the enzymes were thawed and kept on an ice bath for no more than 4 h; re-freezing was not allowed.

### 2.13. Endolysin Activity Turbidimetry Assays

The bacteriolytic activity of HU-Gp82 and HU-Gp84 was determined in a turbidimetry assay as described previously [36], with minor modifications. Briefly, an *E. faecium* FS86 culture (OD590 = 1) was centrifuged at 5000× *g* and 4 °C for 5 min, and then resuspended in a phosphate-buffered LB solution (10 g/L peptone, 5 g/L yeast extract, 10 g/L NaCl, 0.2 g/L KCl, 1.44 g/L Na_2_HPO_4_, 0.24 g/L KH_2_PO_4_ pH 7.5) to an OD590 of approximately 0.4. Next, 20 μL of the purified endolysins were added separately to 380 µL of the FS86 culture in a 48-well microplate (to a final endolysin concentration of 1 µM). *E. faecium* FS86 culture supplemented with 20 μL of Buffer D was used as a control. The microplate was incubated with shaking for 10 h at 35 °C in a FilterMax F5 microplate reader (Molecular Devices, San Jose, CA, USA), with OD595 being measured every 10 min. The experiment was performed as three independent trials. The results were visualized in GraphPad Prism 8.4.3 [16,17] as box-and-whisker plots, with a 5–95% confidence interval.

### 2.14. Endolysins Minimal Active Concentration

The minimum active concentration (MAC) of the endolysins was defined as the concentration that causes a statistically significant decrease in OD595 of a FS86 cell suspension and was determined experimentally using serial fivefold dilutions. The optical density was assessed in a FilterMax F5 microplate reader by the method described above. For subsequent experiments, endolysins were used at concentrations exceeding the MAC by 5–10 times.

### 2.15. pH Optimum for Bacteriolytic Activity and Thermal Stability of Endolysins

To assess the pH-optimum for iF6 bacteriolytic activity, the same buffer solutions were used as in the phage stability assay (described in Section 2.5). Enzyme activity was determined by turbidometry as described above in Section 2.13, with one difference: a set of buffers with different pH values (3–10) were used instead of a phosphate-buffered LB solution. The microplate was incubated with shaking for 1 h at 35 °C, with OD595 being measured every 10 min.

To determine the thermal stability of endolysins, aliquots of endolysins (150 μg/mL) were incubated at various temperatures (20–70 °C) for 1 h. After that, residual enzyme activity was determined by turbidometry, as described above in Section 2.13, during a 1 h incubation at 35 °C with shaking, with OD595 being measured every 10 min.

The enzyme activity of endolysins was shown by a drop in optical density of *E. faecium* FS86 cells under the action of enzymes compared with the control (untreated bacterial cells). The experiments “pH optimum” and “Thermal stability” were performed as five independent trials. The enzyme activity of endolysins was expressed as a percentage, where 100% was taken as the maximum activity of endolysins observed in the experiment. The results were presented as a box-and-whisker diagram, with a 5–95% confidence interval. The results were visualized in GraphPad Prism 8.4.3 [16,17].

### 2.16. Lytic Activity Spectrum of Endolysins

The lytic activity spectrum of endolysins was studied by turbidimetric analysis. Cultures of 26 *Enterococcus* strains and 3 strains from other genera of bacteria (Appendix A) grown to OD590 of 0.4 were used as substrates. The cells were centrifuged, and the pellets were resuspended in PBS buffer (pH 7.5) to the original volume. Aliquots of 380 µL of each cell suspension were transferred to a 48-well microplate, and 20 μL of endolysin preparations or buffer D (as a negative control) were added. The plate was transferred to a FilterMax F5 microplate reader. After intensive mixing, the plate was incubated for an hour, with OD595 being measured every 10 min. The experiment was performed as three independent trials.

### 2.17. Statistical Analysis

Statistical analysis of experimental data was performed using GraphPad Prism 8.4.3 [16,17]. A value of *p* ≤ 0.05 was considered statistically significant.

To evaluate the significance of differences between the concentrations of phages in the “*Thermal and pH stability test*” experiment and the endolysin activities in the “*pH optimum of bacteriolytic activity and thermal stability of endolysins*” experiment, as well as to analyze the statistical differences between the control and tested samples in the experiment “*Spectrum of lytic activity of endolysins*”, a one-way ANOVA with repeated measures was used.

To evaluate the statistical differences in the growth curves of enterococci upon phage infections at different MOIs in the “*Killing assay*”, as well as to analyze the statistical differences between the growth curves of control and tested samples in “*Endolysin activity turbidimetry assays*”, a two-way ANOVA with repeated measures was used.

### 2.18. Accession Number

The nucleotide sequence of the iF6 phage was deposited into the GenBank database under accession number MT909815. The raw datasets can be accessed in the Sequence Read Archive under the accession number SRA SRR12487495.

## 3. Results

### 3.1. Isolation, Host Range and Morphological Characterization of iF6

The *Enterococcus* phage iF6 was isolated from the commercial phage cocktail “Sextaphag^®^”. The phage produced clear plaques from 0.2 to 0.8 mm in diameter on the lawn of the host strain *E. faecium* FS86 (Figure 1a). Host range analysis revealed that iF6 was able to form plaques on the lawns of 9 out of 26 *Enterococcus* strains (35%) and did not produce plaques on the lawns of 3 strains from other genera of bacteria (Appendix A). Thus, the phage is specific to the *E. faecium* species, with the exception of a single *E. thailandicus* strain which it is also capable of infecting.

According to the morphological analysis, iF6 possesses an icosahedral non-elongated capsid of approximately 90.1 ± 2.5 nm in diameter attached to a long, contractile, nonflexible tail of approximately 195.2 ± 8.1 nm in length (including the baseplate structure). The total virion length is 285.2 ± 10.5 nm. The tail ends with a baseplate structure with tail fibers. Thus, iF6 has the characteristic morphological features of the myovirus morphotype (Figure 1b).

### 3.2. Thermal and pH Stability Tests

Phage stability assessment provides important information for storage, transfer and downstream experiments with the phage. Figure 2a shows that iF6 was highly stable at temperatures up to 45 °C, since phage titers were similar to those in the control sample incubated at 5 °C (ANOVA, *p* > 0.05). One-hour incubation at 55 °C and 65 °C reduced the phage titer by a factor of 10 and 1000, respectively (ANOVA, *p* < 0.05). No phage particles were able to survive after one-hour incubation at 75 °C and higher temperatures (ANOVA, *p* < 0.05).

The iF6 bacteriophage was stable at pH 7. During incubation at pH 6 and 8–9, the number of active viral particles decreased by approximately a factor of 100 (Figure 2b) compared with a sample at pH 7 (ANOVA, *p* < 0.05). The phage is not able to survive under highly acidic conditions, as no plaques were detected after incubation at pH 4 and lower pH values (Figure 2b) (ANOVA, *p* < 0.05).

### 3.3. Killing Assay

The lytic activity of *Enterococcus* phage iF6 against *E. faecium* FS86 was assessed at different MOI values (0.1, 1, 3, 5 and 10) by comparing the growth curves of infected and uninfected (control) cultures. The experiment was carried out with cultures of *E. faecium* FS86 in the exponential and stationary phases of growth (Figure 3a,b). A correlation was observed between the initial concentration of iF6 and the inhibitory effect of the phage on bacterial growth, which rose as MOI increased (ANOVA, *p* < 0.001).

In order to evaluate the effect of Ca^2+^ and Mg^2+^ concentrations on the killing activity of iF6, CaCl_2_ and/or MgCl_2_ were added to an *E. faecium* FS86 culture infected with iF6 at an MOI of 10. As shown in Figure 3c,d, the optical density of the bacterial cultures in the presence of Ca^2+^ and/or Mg^2+^ ions decreased similarly to the control culture (without Ca^2+^ and Mg^2+^) (ANOVA, *p* > 0.05). Thus, the presence of Ca^2+^ and Mg^2+^ ions is not essential for the iF6 infection process.

### 3.4. Adsorption Assay and One-Step Growth Curve

The adsorption rate of the iF6 phage was determined by adsorption assay. The adsorption curve in Figure 4a shows that about 90% of the phage particles attach to the host cells within 1 min.

To determine the growth kinetics of the iF6 phage, a one-step growth curve assay was performed, and the main parameters of phage infection were determined. The resulting phage growth curves are shown in Figure 4b. The latent period (virion maturation) of iF6 is about 60 min and the duration of the rise period is 30 min. The average burst size of the phage is 31 ± 1 PFU per infected cell.

### 3.5. General Genome Organization of iF6

Genome sequencing and annotation revealed that the iF6 genomic DNA is a 156,592 bp linear double-stranded DNA with a GC-content of 37.1%. The genomic DNA contains 193 predicted protein-coding sequences (CDS) (Appendix A), 22 predicted tRNA genes, and 1 predicted tmRNA gene (Appendix A). Of the 193 predicted CDSs, 86 (44.56%) were functionally assigned using BLASTp (NCBI) [25] and HHpred [26]. The phage has a relatively large genome with several gene modules, as shown in Figure 5.

*DNA packaging and structural genes*. Two proteins were predicted to relate to phage DNA packaging, both of which were annotated as large terminase subunits: CDS75 (locus_tag iF6_75, protein_id QNL29434.1) and CDS78 (iF6_78, QNL29437.1). It appears that only CDS78 functions as the large terminase, since CDS75 is very short (95 amino acids) and most likely resulted from the insertion of two genes coding for VSR (very short patch repair) endonucleases (iF6_76, QNL29435.1 and iF6_77, QNL29436.1). Four proteins were predicted to be involved in phage capsid assembly. Among them are three proteins that make up the mature viral capsids—major capsid protein (in Figure 5 abbreviated as MCP; iF6_89, QNL29448.1), portal protein (iF6_86, QNL29445.1) and head completion protein (HCP; iF6_91, QNL29450.1), as well as a phage prohead protease (iF6_87, QNL29446.1) required for capsid morphogenesis (maturation) [37]. Eleven protein products were predicted to participate in phage tail assembly. These include one putative tail assembly chaperone (putative TAC; iF6_99, QNL29458.1), facilitating the assembly process, and ten structural proteins constituting the mature tail structure: tail sheath protein (TSP; iF6_96), tail tube protein (TTP; iF6_97, QNL29456.1), tail tape measure protein (TMP; iF6_101, QNL29460.1), baseplate hub protein (iF6_102, QNL29461.1), putative central tail fiber (iF6_103), tail tube terminator protein (iF6_107, QNL29466.1) and four baseplate assembly proteins (iF6_109, QNL29468.1; iF6_110, QNL29469.1; iF6_111, QNL29470.1; iF6_112, QNL29471.1).

*Lytic genes*. Five proteins have been predicted to be associated with host bacterial cell lysis, including putative hemolysin XhlA (iF6_70, QNL29429.1), N-acetylmuramoyl-L-alanine amidase (iF6_82, QNL29441.1), LysM domain-containing peptidoglycan-binding protein (iF6_83, QNL29442.1), N-acetylmuramoyl-L-alanine amidase (iF6_84, QNL29443.1) and holin (iF6_145, QNL29504.).

*Replication, Repair and Recombination*. The module of replication, repair and recombination-related proteins includes the UvsW-like ATP-dependent DNA helicase (iF6_117, QNL29476.1), DnaB-like replicative helicase (iF6_119, QNL29478.1), DNA repair exonuclease SbcCD-like nuclease subunit (iF6_120, QNL29479.1), DNA repair exonuclease SbcCD-like ATPase subunit (iF6_121, QNL29480.1), DNA primase (iF6_123, QNL29482.1), dUTPase (iF6_124, QNL29483.1), Holliday-junction resolvase (iF6_131, QNL29490.1), 3′–5′ exonuclease and polymerase domain-containing uracil DNA glycosylase family 4 (iF6_137, QNL29496.1), 3′–5′ exonuclease and polymerase domain-containing DNA polymerase I (iF6_139, QNL29498.1), UvsY-like protein (iF6_140, QNL29499.1), Gp32-like single strand DNA-binding protein (iF6_141, QNL29500.1), RecA-like protein (iF6_142, QNL29501.1), DnaJ-type Zn-binding domain-containing RecJ-like exonuclease (iF6_150, QNL29509.1), metallophosphatase domain-containing protein (Mre11-like nuclease) (iF6_151, QNL29510.1) and MmcB-like protein (iF6_154, QNL29513.1).

More detailed information about the predicted CDSs is shown in Appendix A.

### 3.6. Comparative Genomics

A BLASTn search was performed against the NCBI nr database using the iF6 genome as the query, which resulted in the identification of sixteen closely related *Enterococcus*-infecting phages: 163 (CAJDKA010000002.1), EFP01 (NC_047796.1), EFDG1 (NC_029009.1), PEf771 (MN241318.1), vB_OCPT_Ben (MN027503.1), EfV12-phi1 (NC_048087.1), 156 (LR031359.1), EFLK1 (NC_029026.1), ECP3 (NC_027335.1), phiEF17H (AP018714.1), phiM1EF22 (AP018715.1), vB_EfaM_Ef2.3 (MK721192.1), phiEF24C (NC_009904.1), vB_EfaM_Ef2.1 (MK693030.1), vB_EfaH_EF1TV (MK268686.1) and PBEF129 (MN854830.2). The complete genomes of the above-listed phages were downloaded from GenBank, and a whole-proteome comparison based on the translated ORFs of the phages was performed using VICTOR (formula D6) [32]. The resulting phylogram illustrates the evolutionary relationship between *Enterococcus* phage iF6 and the closest viruses (Figure 6). As shown in Figure 6, iF6 shares one clade with six phages belonging to the genus *Schiekvirus*. The branching order is consistent with the number of proteins shared by the six most closely related phages with iF6 (Appendix A).

The linear whole-genome comparison diagram in Figure 7 shows tBLASTx pairwise similarities between iF6 and the two most closely related viruses, *Enterococcus* phages 163 and EFP01.

### 3.7. Determination of Packaging Strategy

It is known that different types of bacteriophage chromosome termini result from different DNA replication and packaging strategies. Complete genome sequencing is not always sufficient to elucidate the type and location of phage chromosome termini; therefore, several experimental approaches, such as restriction analysis, have been adopted to accomplish this goal [34].

Genome ends of iF6 were verified by restriction analysis (Figure 8a) and the RAGE method, followed by Sanger sequencing (Figure 8b). The analyses revealed the presence of two long direct terminal repeats (DTRs) at the chromosome ends, each 2108 bp in length. Schematic representation of the phage chromosome is shown in Figure 8c.

### 3.8. Cloning and Characterization of Recombinant Endolysins

Two genes, *iF6_82* and *iF6_84*, annotated as N-acetylmuramoyl-L-alanine amidases, were chosen for the analysis of their bacteriolytic potential. Gp82-H and Gp84-H proteins were successfully produced in an *E. coli* BL21 Star (DE3) pET33 system; however, these aggregated during purification. The precipitates were irreversibly insoluble; even heating in the presence of 1% SDS did not enable protein recovery from the aggregates. Since optimization of expression conditions did not improve the solubility of the proteins, the enzymes were produced as translational fusions with the ubiquitin domain at the N-terminus of the molecules. The ubiquitinylated HU-Gp82 and HU-Gp84 molecules showed sufficient stability in solution and were well expressed (Appendix A).

As shown in Figure 9a,b, both HU-Gp82 and HU-Gp84 enzymes showed bacteriolytic activity against *E. faecium* FS86 cultures (ANOVA, *p* < 0.05). The determined minimum active concentrations (MAC) of the endolysins were 30 and 22 μg/mL for HU-Gp82 and HU-Gp84, respectively. Both enzymes exhibited preferable bacteriolytic activity to bacteriophage iF6: they were active against both logarithmic and stationary cultures of *E. faecium* FS86 (Figure 9).

### 3.9. pH Optimum of Bacteriolytic Activity and Thermal Stability of Endolysins

The pH optima of endolysin bacteriolytic activities significantly differed: the maximum activity of HU-Gp82 was observed at pH 6, while for HU-Gp84, this occurred at pH 8 (Figure 10a,b) (ANOVA, *p* < 0.05). HU-Gp82 retained more than 50% of the maximum activity in pHs ranging from 7 to 9, while HU-Gp84 was active in the pH range of 5–9.

HU-Gp82 lost about half of its activity after incubation at 30 °C for an hour and was almost completely inactivated at 60 °C (Figure 10c) (compared with a sample at 20 °C as a control, ANOVA, *p* < 0.05). HU-Gp84 almost completely retained its bacteriolytic activity even upon incubation at 60 °C (compared with a sample at 20 °C as a control, ANOVA, *p* > 0.05), but it was completely inactivated at 70 °C, and no residual activity was detected (compared with a sample at 20 °C as a control, ANOVA, *p* < 0.05) (Figure 10d).

### 3.10. Spectrum of Lytic Activity of Endolysins

The lytic activity of endolysins was evaluated on 26 *Enterococcus* strains and 3 strains from other genera of bacteria. HU-Gp82 was able to lyse 35% (9 out of 26) of the strains tested, while HU-Gp84 lysed 77% (20 out of 26) of the strains. Whenever the minimal endolysin dose (15 μg/mL HU-Gp82 or 11 μg/mL HU-Gp84) was sufficient to ensure culture lysis, the strain was considered highly susceptible (++) (Appendix A). If a larger dose (30 μg/mL HU-Gp82 or 22 μg/mL HU-Gp84) was required, the strain was marked as less susceptible (+) (Appendix A). When the endolysins had no effect on the culture, the strain was considered resistant (-) (Appendix A).

Thus, the endolysins showed different strain specificities compared with each another and with the iF6 phage (Appendix A). Three non-enterococcal strains tested were resistant to the iF6 phage and to both endolysins (Appendix A).

## 4. Discussion

The *Enterococcus* bacteriophage iF6, which was isolated from the commercial phage cocktail “Sextaphag^®^” and characterized in this work, belongs to *Herelleviridae*, subfamily *Brockvirinae*, genus *Schiekvirus* (Figure 6). According to the morphological analysis, iF6 has a long, contractile, nonflexible tail and a baseplate structure with tail fibers. The virion structure is typical of phages with a myovirus morphotype (Figure 1). The chromosome of iF6 is a dsDNA molecule with a length of 156,592 bp and two direct terminal repeats of 2108 bp each (Figure 5 and Figure 8c). Similar organization of both virions and genomes has been described in phages infecting enterococci [38,39,40,41,42,43].

According to the phylogenetic analysis, the most closely related phage to iF6 is *Enterococcus* phage 163 (vB_EfmH_163) [42] (Figure 6). The vB_EfmH_163 and iF6 phages have 86% whole-genome identity (Appendix A). The number of shared proteins between these phages is 159 (42%) (Figure 7, Appendix A).

The thermal stability of iF6 (Figure 2a) and vB_EfmH_163 is similar; in contrast, the pH stability of these phages differs significantly. The vB_EfmH_163 phage is stable in pH ranging from 2 to 9, while iF6 has a narrow range of pH stability (near pH 7) (Figure 2b). The difference in the pH stability of the phages may be related to the conditions under which the experiments were carried out, in particular, PBS buffers with pH 2–9, in which the vB_EfmH_163 phage was incubated [42].

iF6 showed excellent adsorption kinetics as almost 90% of the virions were adsorbed within 1 min (Figure 4a). At the same time, the number of virions produced in one cell was found to be low. The average burst size of iF6 was 31 ± 1 PFU per infected cell, while the latent period was about 60 min, and the duration of the rise period was 30 min (Figure 5b). Of all representatives of *Shiekvirus* and *Kochikohdavirus*, bacteriophages have the largest progeny: vB_EfmH_163 (155 PFU per infected cell [42]), phiEF24C (110-120 PFU per infected cell [38]), and EfV12-phi1 (135 PFU per infected cell [44]), notwithstanding the tiny dimensions of their plaques—only about 0.3 mm in diameter [44,45]. The limited burst size of enterococcal phages belonging to these two genera can be explained by the large size of their virions (~0.285 µm from the top of the head to the baseplate), and the small size of the host cells (~0.8 µm) [46]. Uchiyama and co-authors described a spontaneous mutant of the phiEF24C phage, phiEF24C-P2 (AB609718), which formed 4–5 mm plaques with halos [45]. It seems that the plaque size of phiEF24C-P2 did not correlate with its burst size because the charge-reverse mutation in ORF31 (99% coverage, 71.66% identity with iF6_103, QNL29462.1) did not lead to an increase in phiEF24C-P2 progeny but increased the phage adsorption rate on the two tested cultures of *E. faecalis* [45]. Apparently, the size of the phage progeny does not correlate with the size of the plaques in iF6-related phages. The EFDG1, KFKL1 [39] and EF1TV [47] phages, as well as iF6, show low progeny values, several times smaller than vB_EfmH_163, phiEF24C or EfV12-phi1. Small progeny size can be a result of a suboptimal phage development within the host bacterium. It is possible that iF6 development is compromised upon DNA replication and/or DNA packaging into capsids, as the micrographs show a large number of virions with defective capsids (Appendix A). Defective capsids can also be observed in micrographs of the EFDG1 phage [48].

A study of the lytic activity of the *Enterococcus* iF6 phage against *E. faecium* FS86 was conducted, which showed that iF6 can inhibit the growth of an FS86 culture, both in the exponential and in the stationary growth phase (Figure 3a,b, respectively). In addition, the presence of Ca^2+^ and Mg^2+^ did not affect the killing activity of iF6 in either growth phase. Thereby, phage preparations or phage cocktails based on the iF6 phage do not require additional bivalent ions (such as calcium and magnesium ions) to ensure the effectiveness of phage therapy.

The iF6 phage has a highly similar genome organization to the vB_EfmH_163 phage. The restriction and RAGE analysis revealed the presence of DTRs at the ends of the iF6 chromosome, each being 2108 bp in length (Figure 8a,b). In addition, no virulence factors, such as toxin-encoding genes or antibiotic-resistance genes, were detected in the iF6 genome, which makes a case for using iF6 in phage therapy.

Members of *Schiekvirus* and *Kochikohdavirus* have established themselves as promising candidates for combating antibiotic-resistant enterococci. Mice treated with phiEF24C were cured of vancomycin-resistant *E. faecalis* bacteremia [49]. Bacteriophage EFDG1 (NC_029009.1) effectively killed *E. faecalis* in an ex vivo root canal infection model [48]. The EFKL1 phage eliminated antibiotic- and phage-resistant *E. faecalis* in vitro and in a fibrin clot model [39]. A mixture of EFKL1 and EFDG1 killed *E. faecalis* in an in vivo model of rat root canal infection [50]. A cocktail of three phages, phiEF7H, phiEF14H1 and phiEF19G, closely related members of the genus *Kochikohdavirus*, was highly effective against endophthalmitis caused by vancomycin-resistant *E. faecalis* in both in vivo and in vitro models [40]. The EF1TV phage was able to eradicate biofilms formed by clinical isolates of *E. faecalis* [47]. The vB_EfmH_163 phage reduced mortality caused by infection with an *E. faecium* vanR clinical isolate in a *Galleria mellonella* animal model [42]. Thus, iF6, as well as other members of *Schiekvirus* and *Kochikohdavirus*, can be used as antibacterial agents against antibiotic-resistant enterococci infections.

However, it should be taken into account that not only phages may be of therapeutic interest as potential antibacterial agents, but also their lytic enzymes, such as endolysins. In the last few decades, a range of iF6-like bacteriophages of the genera *Schiekvirus* and *Kochikohdavirus* have been sequenced and described, and the properties of some of their endolysins have been studied.

Similar to most endolysins of phages infecting Gram-positive bacteria, iF6 endolysins consist of two domains: an enzymatically active domain (EAD) and a cell-wall-binding domain (CBD). The N-terminal part of Gp82, harboring an EAD (1-187aa, i.e., coverage 44%), is 84.13% identical to the previously studied endolysins Lys170 (BBE37284) of the F170/08 phage [41] and ORF9 (BAF81277) of the *E. faecalis* phage phiEF24C [38]. The peptidoglycan cleavage site of the phiEF24C ORF9 endolysin (YP_001504118) was located experimentally, thus confirming the N-acetylmuramoyl-L-alanine amidase activity of the protein [51].

Most endolysin molecules are monomers; however, Lys170 (BBE37284) of bacteriophage F170/08 is a heterotetrametric complex consisting of one full-length monomer chain and three short chains harboring only carbohydrate-binding domains (CBD), which are synthesized from an additional ribosome-binding site in the *lys*170 mRNA [52]. The CBD of Gp82 does not demonstrate much similarity to the CBD of Lys170. To date, there are no experimental data on the CBD of Gp82 and its homologs. It can be assumed that the Gp82 CBD can have a complex organization, similarly to Lys170, taking into account the aggregation occurring during its purification in a heterologous system.

Endolysin Gp84 demonstrates a high similarity to PlyV12 (AAT01859) of the bacteriophage EfV12-phi1 (100% coverage, 97% identity). PlyV12 has a broad-spectrum bacteriolytic activity against several types of enterococci, streptococci and staphylococci [53]. PlyV12 is also highly identical (100% coverage, 99% identity) to AVV19_gp120 (YP_009218318.1) of the bacteriophage EFDG1 (NC_029009.1), which effectively destroyed *E. faecalis* in an ex vivo model of root canal infection [48]. In one of the previous studies, the P10N-V12C chimeric lysin was constructed, which consisted of the PlyV12 CBD domain and the PlyP10N EAD domain. Due to the broad activity spectrum of the PlyV12 CBD, the P10N-V12C enzyme was able to lyse both enterococci and staphylococci. Interestingly, P10N-V12C may be species-selective for enterococci because it was unable to lyse any of the five *E. faecium* strains tested but destroyed 10 out of 10 *E. faecalis* strains [54]. Thus, endolysins are of great interest to the scientific community due to their specificity and mechanism of action, as well as the potential for engineering and lack of resistance mechanisms.

It is well known that endolysins often have a broader range of susceptible bacterial strains than phages. The HU-Gp84 endolysin was able to lyse 20 (77%) out of 26 *Enterococcus* strains, in contrast to the HU-Gp82 endolysin and the iF6 phage, each of which lysed 9 strains (Appendix A). Thus, HU-Gp84 and its homologs can be used as therapeutic agents against antibiotic-resistant enterococci.

## 5. Conclusions

*Schiekvirus* and *Kochikohdavirus* enterococcal phages are found in sewage globally. They have been isolated in Canada, Japan, Israel, Italy and Russia. It appears that iF6-like enterococcal phages are a promising platform for the search, selection, and development of new candidates for phage therapy. The application of these phages, including the *Schiekvirus* iF6 described in this study, as components of bacteriophage cocktails for therapeutic purposes seems to be warranted. However, more extensive investigation is required for a better understanding of the phages’ interaction with and adaptation to the bacterial host, the accumulation of mutations and the possibility of horizontal gene transfer.

## Figures and Tables

**Figure 1 viruses-15-00767-f001:**
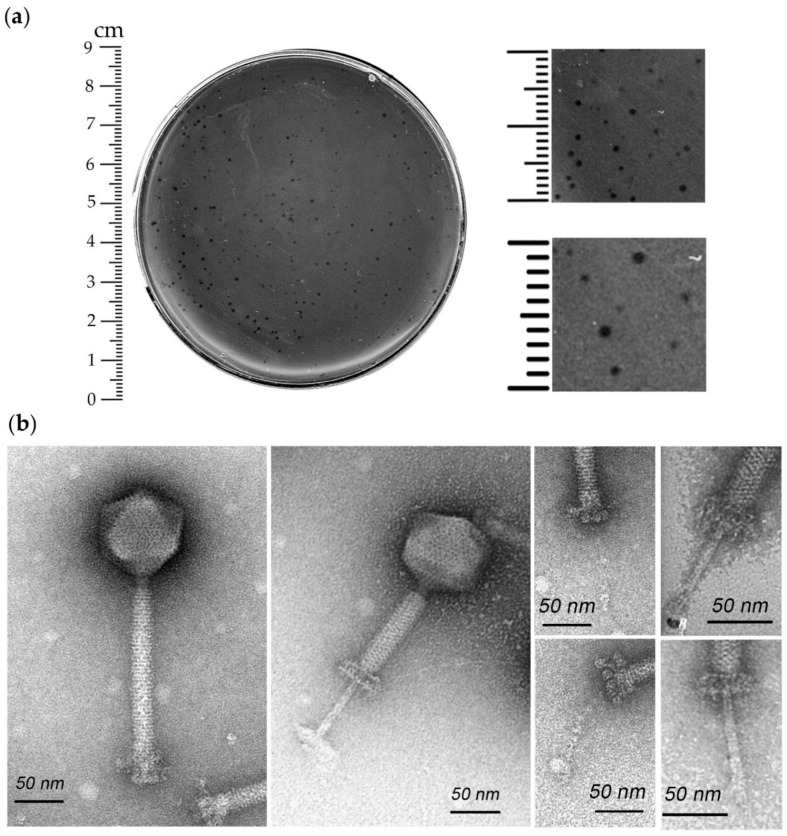
(**a**) Morphology of *Enterococcus* phage iF6 plaques on a lawn of the *E. faecium* FS86 strain. (**b**) Transmission electron microscopy of *Enterococcus* phage iF6 virions with uncontracted (left virion) and contracted (right virion) tails. Virions are negatively stained with 1% (*w*/*v*) uranyl acetate. Scale bar is 50 nm.

**Figure 2 viruses-15-00767-f002:**
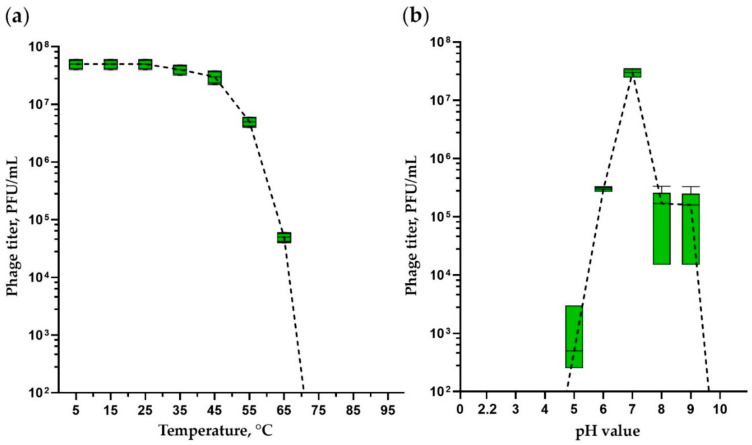
Thermal and pH stability of *Enterococcus* phage iF6. (**a**) Phage titers after 1 h incubation at temperatures ranging from 5 to 95 °C. (**b**) Phage titers after 1 h incubation at pH values ranging from 2.2 to 10. The graphs were created in GraphPad Prism 8.4.3 with a 95% confidence interval. Five independent trials were performed for the experiments.

**Figure 3 viruses-15-00767-f003:**
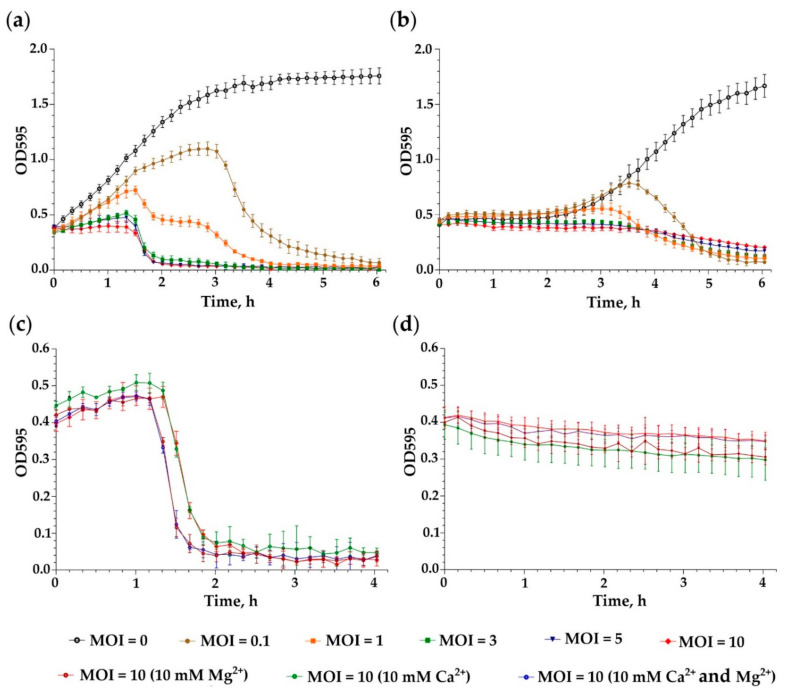
The growth curves of FS86 upon iF6 infection at different MOIs with additional Ca^2+^ or/and Mg^2+^. Growth kinetics of *E. faecium* FS86 upon infection at various MOI values in (**a**) the exponential and (**b**) the stationary growth phases. An uninfected culture was used as a control. Calcium and magnesium effects on the killing activity of iF6 in (**c**) the exponential and (**d**) the stationary phases of growth. The graphs were created with GraphPad Prism 8.4.3. Error bars represent the standard deviation of the means from three independent trials.

**Figure 4 viruses-15-00767-f004:**
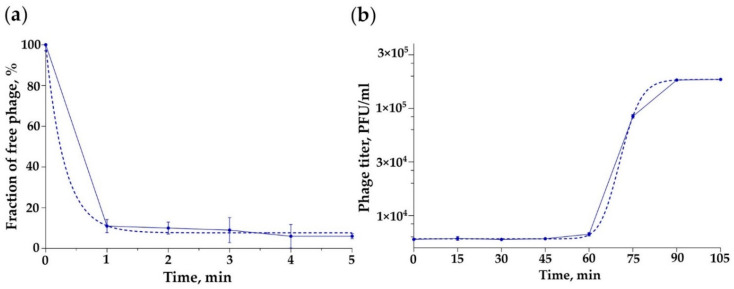
Phage infection parameters. (**a**) Adsorption and (**b**) one-step growth curve of *Enterococcus* phage iF6. The graphs were created with GraphPad Prism 8.4.3. Error bars represent the standard deviation for average values from three independent trials. The dotted line marks the interpolation according to (**a**) the exponential model and (**b**) the sigmoid model.

**Figure 5 viruses-15-00767-f005:**
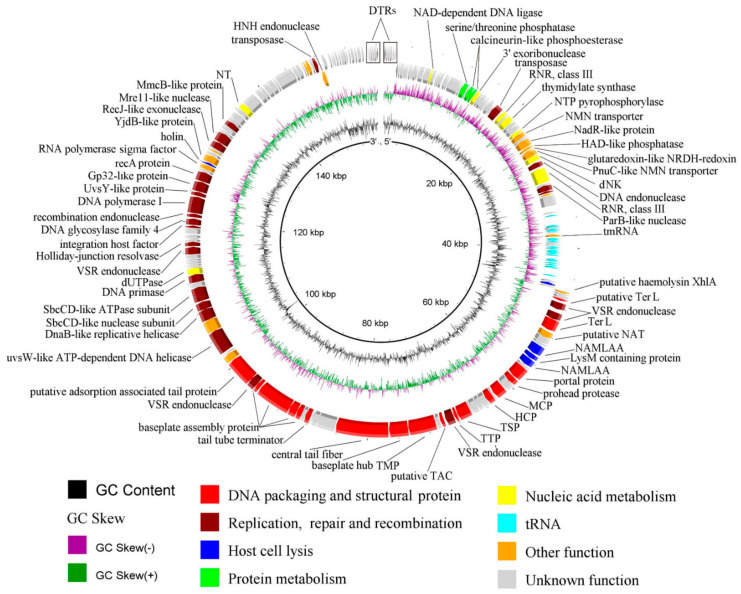
The *Enterococcus* phage iF6 chromosome map. Functionally assigned ORFs are highlighted according to their general functions (see the color scheme at the bottom). Abbreviations: DTRs—direct terminal repeats; RNR—ribonucleotide reductase; NTP pyrophosphorylase—nucleoside triphosphate pyrophosphorylase; NMN transporter—nicotinamide mononucleotide transporter; dNK—deoxyribonucleoside kinase; TerL—terminase, large subunit; NAMLAA—N-acetylmuramoyl-L-alanine amidase; MCP—major capsid protein; HCP—head completion protein; TSP—tail sheath protein; TTP—tail tube protein; VSR endonuclease—very short patch repair endonuclease; TAC—tail assembly chaperone; TMP—tape measure protein; NT—nucleotidyltransferase.

**Figure 6 viruses-15-00767-f006:**
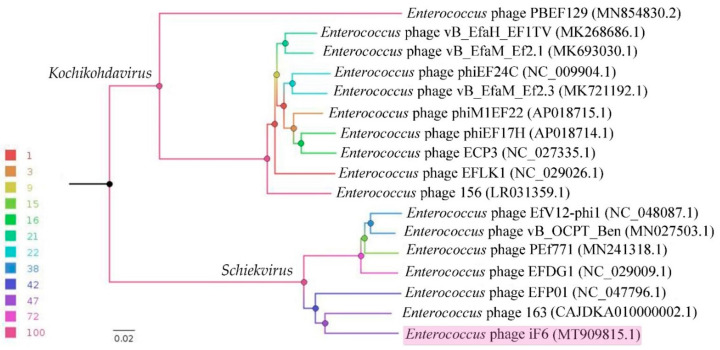
Genome-BLAST Distance Phylogeny (GBDP) phylogram based on the amino-acid sequences of the whole phage proteomes and inferred using formula D6. The nodes are colored according to the legend representing GBDP pseudo-bootstrap support values from 100 replications. The branch lengths are scaled in terms of the GBDP distance (formula D6). The *Enterococcus* phage iF6 is highlighted in light purple.

**Figure 7 viruses-15-00767-f007:**
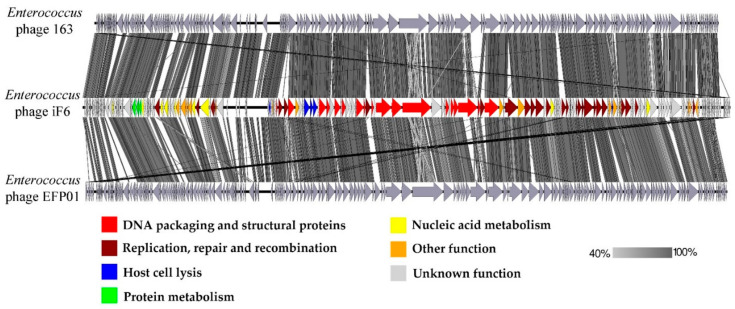
The pairwise whole-genome tBLASTx comparison of iF6 and the most closely related phages. The iF6 ORFs are colored according to their general function (see the color scheme at the bottom). Gray areas between the genome maps indicate the level of identity in the range of 40% to 100% (see the gradient scheme on the right).

**Figure 8 viruses-15-00767-f008:**
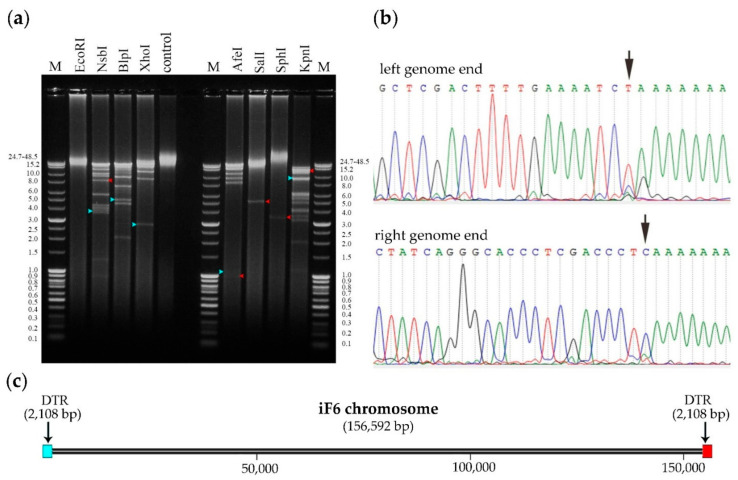
Determination of packaging mechanism and genome termini. (**a**) Restriction analysis of iF6 DNA. M—molecular weight markers; red and light blue arrows indicate fragments containing the right and left ends of the iF6 chromosome, respectively. The original full-length gel is presented in Appendix A. (**b**) Sequencing chromatograms of the terminal regions of the PCR product sequences obtained with RAGE for the left and right ends. (**c**) Schematic representation of the iF6 phage chromosome.

**Figure 9 viruses-15-00767-f009:**
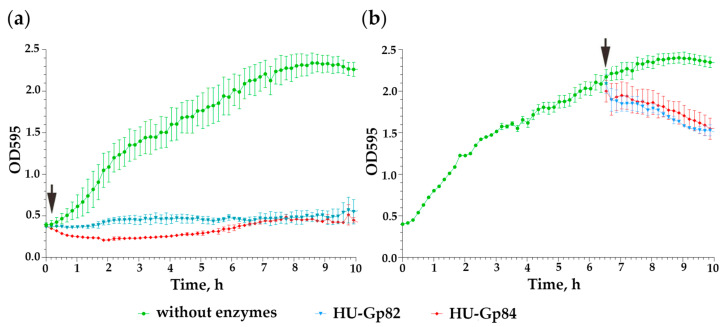
Effect of HU-Gp82 and HU-Gp84 endolysins on the growth kinetics of *E. faecium* FS86. Enzymes were added to the cell suspension in the early exponential (**a**) and stationary (**b**) phases of growth. An uninfected culture of *E. faecium* FS86 was used as a control. The black arrows indicate the addition of endolysins. The graphs were created using GraphPad Prism 8.4.3. Error bars represent the standard deviation of the means from three independent trials.

**Figure 10 viruses-15-00767-f010:**
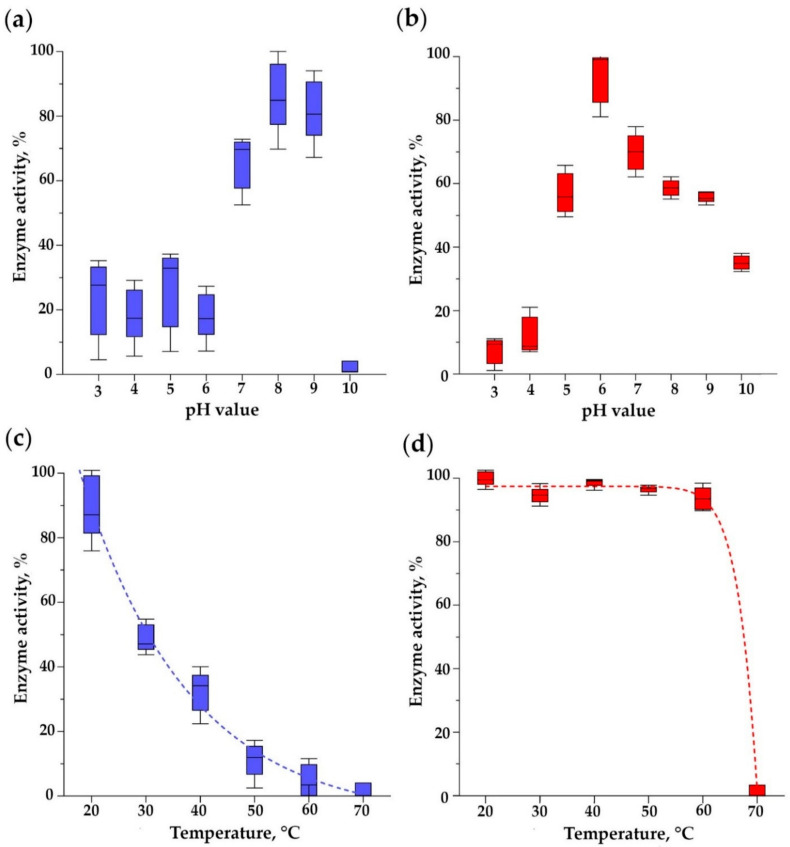
The bacteriolytic activity of (**a**) HU-Gp82 and (**b**) HU-Gp84 endolysins at different pH values at 35 °C. The residual activity of (**c**) HU-Gp82 and (**d**) HU-Gp84 endolysins after 1 h incubation at different temperatures. The graphs were created with GraphPad Prism 8.4.3 with a 95% confidence interval. Five independent trials were performed in the experiments.

## Data Availability

The genome sequence for iF6 phage was deposited into the GenBank database under accession number MT909815.1. The raw datasets can be accessed at the Sequence Read Archive under the accession number SRA SRR12487495.

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
