# Peer review of "Bacteriolytic Potential of Enterococcus Phage iF6 Isolated from “Sextaphag®” Therapeutic Phage Cocktail and Properties of Its Endolysins, Gp82 and Gp84"

_viruses, 2023, doi:10.3390/v15030767_

Round 1

Reviewer 1 Report

The research article "Bacteriolytic Potential of Enterococcus Phage iF6 Isolated from "Sextaphag®" Therapeutic Phage Cocktail and Properties of its Endolysins, iF6_82 and iF6_84 " by Buzikov et al. described the virulent bacteriophage iF6 and the properties of two of its endolysins. This study is needy and performed enough. But there are some major queries that need to be addressed by the authors.

Major:

1.     The authors claimed that Phage iF6 was isolated from commercial phage cocktail "Sextaphag®", and commercial cocktail phages are made of what kinds of phages? Which host bacteria are the main targets? How can you be sure that the isolated phage is different from the original ingredient of the cocktail at first?

2.     The shape of plaque in Figure 1A is not very uniform? 

3.     Why to evaluate the effect of Ca 2+ and Mg 2+ concentration on the killing activity

of iF6? How about Zn2+?

4.     Line 390-397, By comparing the genome, what is the homology of this phage and its similar phage? The author should list it.

5.     How about the bactericidal ability of endolysins? Compare that to phages!

6.     The author didn't do animal studies to evaluate the efficacy of phages or endolysins, but is it possible to evaluate the bactericidal efficacy of phages or endolysins in cells?

Author Response

“The research article "Bacteriolytic Potential of Enterococcus Phage iF6 Isolated from "Sextaphag®" Therapeutic Phage Cocktail and Properties of its Endolysins, iF6_82 and iF6_84 " by Buzikov et al. described the virulent bacteriophage iF6 and the properties of two of its endolysins. This study is needy and performed enough. But there are some major queries that need to be addressed by the authors.”

Response:

Thank you for appreciating our work. We have carefully considered all the suggestions and comments, and made revisions accordingly. All the changes made to the manuscript are marked up using the “Track Changes” in the word file of the revised manuscript. Please see our point-by-point response to the comments and concerns below:

Point 1:

“The authors claimed that Phage iF6 was isolated from commercial phage cocktail "Sextaphag®", and commercial cocktail phages are made of what kinds of phages? Which host bacteria are the main targets? How can you be sure that the isolated phage is different from the original ingredient of the cocktail at first?”

Response 1:

Unfortunately, the final qualitative formulation (composition) of the commercial phage cocktail "Sextaphag®" is not fully disclosed by the manufacturer. In general, pharmaceutical development information is commercially confidential. This includes detailed data concerning active substance, formulation and manufacturing, test procedures and validation. The "Sextaphag®" manufacturer indicated a list of bacterial species against which this phage cocktail is effective: Staphylococcus spp., Streptococcus spp., Proteus vulgaris, Proteus mirabilis, Pseudomonas aeruginosa, Klebsiella pneumoniae, Escherichia coli.

We are confident that the bacteriophage iF6 isolated by us is one of the bacteriophages of the "Sextaphag®", since all microbiological procedures such as isolation, extraction and growth of the bacteriophage iF6 preparation were carried out at a proper professional level, which excludes the possibility of contamination from external sources. However, we cannot rule out the fact that the isolated iF6 may have acquired some mutations (particularly point mutations) compared to the original phage in the commercial product. In addition, it is well known that bacteriophages can also acquire mutations during large-scale industrial cultivation. Thus, the phage cocktail (the genetic component of the phages) may differ from lot to lot. The ability of a candidate-phage for mutagenesis should be assessed by the manufacturer when developing a phage cocktail and should not exceed a critical level and, as a result, affect the properties of this phage. Based on the taxonomic criteria established by the International Committee on Taxonomy of Viruses (ICTV), the criterion for species is more 95% DNA sequence identity. (Dann Turner et al., 2021 [https://doi.org/10.3390/v13030506].). Therefore, 5% genetic variability (no effect on phage properties) for a candidate-phage is acceptable and does not change the phage's species affiliation.

Thus, taking into account all of the above, we are fully confident that the isolated iF6 phage is the same species as one of the original phages in the "Sextaphag®" preparation.

Point 2:

“The shape of plaque in Figure 1A is not very uniform?”

Response 2:

Indeed, Figure 1 shows that iF6 plaque morphology may differ slightly on a lawn of E. faecium FS86 strain: the size of the phage plaque can vary from 0.2 to 0.8 mm. As described in the section 2 “Materials and Methods”, subsection 2.2.“Phage isolation, purification and propagation”: in order to exclude the presence of other phages with morphologically differ or identical plaques, the extraction-titration cycles were repeated three times (three consecutive passages were carried out). DNA from the purified phage preparation was isolated and sequenced. according to the results of sequencing, the presence of other phages in the purified iF6 phage sample was not detected.

The slight variability in plaque morphology may be related to phage-bacteria interactions.

Point 3:

“Why to evaluate the effect of Ca 2+ and Mg 2+ concentration on the killing activity of iF6? How about Zn2+?”

Response 3:

It is well known that the adsorption efficiency of some bacteriophages, as well as the activity of their enzymes, depends on the presence of magnesium and/or calcium ions (rarely strontium and barium ions) [Paranchych, W. (1966). Stages in phage R17 infection: the role of divalent cations. Virology, 28(1), 90-99. https://doi.org/10.1016/0042-6822(66)90309-6]. The protocol described in section in the section 2 “Materials and Methods”, subsection 2.6.“Killing assay and the effect of Ca2+ and Mg2+” is very standard and is commonly used in phage research. Strontium and barium ions are used in medicines to a limited extent, so assessing their influence is of no practical importance.

Regarding your question about the effect of zinc ions on the lytic activity of the iF6 phage and its endolysins: the zinc ion belongs to another group of ions (Zn2+ is in the same group-II in periodic table of chemical elements as cadmium and mercury ions).

We evaluated the effect of Zn2+ ions on the lytic activity of the iF6 phage. The experiment was carried out in three independent trials. At a zinc ions concentration of 10 mM (at the same concentration as calcium and / or magnesium ions in the experiments carried out earlier), it had an acute toxic effect on bacterial cells (Figure 1) (compared with sample without Zn2+, Ca2+ and Mg2+, ANOVA p<0.05). At a zinc concentration of 1 mM, Zn2+ ions inhibited the growth of bacterial cells, which led to a 20-30-minute delay in the growth kinetics of the bacterial culture E. faecium FS86 compared to the control culture (no addition of zinc ions) (Figure 1) (compared with sample without Zn2+, Ca2+ and Mg2+, ANOVA p<0.05). At a zinc concentration of 0.1 mM, statistically significant differences (ANOVA p>0.05) were not found.

Figure 1. The effect of different ions on the decrease in optical density of E. faecium FS 86 upon infection with iF6 at MOI=10.

Point 4:

“Line 390-397, By comparing the genome, what is the homology of this phage and its similar phage? The author should list it.”

Response 4:

As you rightly noted, some of the related phages to the iF6 phage in Figure 6 do not have information about BLASTn genome identity to iF6 in the main text of the manuscript, since we believe that there is no essential need to specify the BLASTn identity parameters for all listed phage genomes in lines 390-397. In our opinion, addition of genome identity values for each listed phage genome will complicate reading the manuscript. Therefore, we preferred to leave the genomes identity values for viruses that share the same clade with iF6, in the Supplementary Information, Table S4.

Point 5:

“How about the bactericidal ability of endolysins? Compare that to phages!”

Response 5:

Thank you for your comment. The information about the comparison of the lytic spectrum of the enterococcal phage iF6 and its endolysins have listed in Supplementary Information, Table S1.

Also, we have added information about the comparison of the bactericidal ability of HU-Gp82 and HU-Gp84 endolysins and iF6 phage in Discussion section of manuscript.

Point 6:

“The author didn't do animal studies to evaluate the efficacy of phages or endolysins, but is it possible to evaluate the bactericidal efficacy of phages or endolysins in cells?”

Response6:

Undoubtedly, we agree that test procedures and the validation of phage-based drugs is an important step in a drug development. However, the aim of this work is a study of biochemical and antibacterial properties of the iF6 bacteriophage and its bacteriolytic molecules. As a result of our studies Gp84 molecule show good bacteriolytic properties. We interested in study Gp84 and on an eucaryotic/animal infection model in future work, which can be a subject for next publications from our laboratory.

In addition, we updated all the figures corrected in accordance with the requirements of the Viruses journal:

Figure 1: The designations of panels A and B have been changed to (a) (b), a “cm” mark has been added next to the ruler.

Figure 2: The vertical axes has been adjusted for both panels. The vertical axes on panels (a) and (b) have been renamed to "Phage titer, PFU/ml". Both vertical axes have been set to the same scale.

Figure 3: The font size has been increased. The color scheme of the lines has been rearranged to be more readable.

Figure 4: The font size has been increased. For panel (b), the vertical axis has been renamed from “PFU/mL” to “Phage titer, PFU/mL”.

Figure 5: no changes.

Figure 6: All Latin names have been italicized.

Figure 7: Changed the red color legend in the color scheme from “DNA packaging and structural protein” to “DNA packaging and structural proteins”

Figure/ 8: The designations of panels A and B have been changed to (a) (b) and (c), text front changed to Palatino Linotype.

Figure 9: Arrow marks have been added to indicate when endolysins were added. The line label color scheme for lysis curves has been simplified. The font size has been increased.

Figure 10: The font size has been increased. The name of the horizontal axes has been changed from "pH" to "pH value" in panels (a) and (b).

Reviewer 2 Report

Global comments

The work presented by Buzikov and co-workers describes the characteristic of iF6 phage against Enterococcus isolated from commercial product  Sextaphag® and its lysins and their antibacterial activity.

The paper presents an interesting study and the results can promote the idea of developing lysins usage as antimicrobials. The introduction properly situates the subject of study, and the material and methods are adequately documented., but need statistical analysis. Results are also correctly exposed, but some of them need statistical analysis. However, the lysins against Gram-positive bacteria are well characterized and their antibacterial activity is detailed described. Therefore results obtained in this study should be discussed with the existing literature background.

I include all my comments, and I hope the authors are willing to take the time to make this work a solid publication that phage community can refer to.

I refer to line number kindly provided by the authors:

Materials and Methods section:

Please give the manufacturer of each reagent or medium used in this study.

Line 116: Enterococcus should be italicized

Line 123 – 125; 294-250: Why author used different buffers, with and without glycine? The presence/absence of glycine could influence phage pH stability. In my opinion, the authors should adjust with 1 M NaOH or HCl the phage cocktail to yield a pH range of 2–13, according to Lu and colleagues (2020).

Lines 262-266: This paragraph should be written in the results section.

Some analyses need a statistical analysis such as temperature, pH stability, between MOI and between endolysins. With no statistical analysis, we can not talk about differences only about the trend.

Results Section:

Lines 288-289: 1 hour of incubation is insufficient to conclude how different temperatures can influence phage activity. We can talk only about stability for 1 hour. I suggest checking 1-week storage at 4, 20 and 37°C to determine stability in the human body temperature and storage conditions.

Lines 388: Please describe the content of genes responsible for lytic/lysogenic cycle – this is the most important criterium for phage selection for therapeutic purposes, as well as known genes encoding toxins, antibiotic resistance and virulence or superinfection immunity protein. The figure is insufficient to interpret by the reader.

Lines 442-455: Authors should add SDS-PAGE gels pictures to show aggregates and protein products of expression indicating their proper weight.

Line 485: Please add citations.

In the Results section authors describe endolysins HUgp82 and HUgp84, while in the Discussion section the names iF6_82 and iF6_84. It is not clear if we talk about the same proteins. Please use the same name everywhere.

Discussion:

I suggest rewriting the discussion and starting with the overall characteristic of iF6 phage and then describing details concerning endolysins. It will build a more interesting story. Authors should discuss the great potential of HUgp84 in Enterococcus combating and its resistance to harsh conditions. Please compare it with other papers where authors indicate a higher ability of lysins than phages to be an antibacterial tool even if they are against other pathogens than Enterococcus. 

Author Response

Response to Reviewer 2

“The work presented by Buzikov and co-workers describes the characteristic of iF6 phage against Enterococcus isolated from commercial product Sextaphag® and its lysins and their antibacterial activity.

The paper presents an interesting study and the results can promote the idea of developing lysins usage as antimicrobials. The introduction properly situates the subject of study, and the material and methods are adequately documented., but need statistical analysis. Results are also correctly exposed, but some of them need statistical analysis. However, the lysins against Gram-positive bacteria are well characterized and their antibacterial activity is detailed described. Therefore results obtained in this study should be discussed with the existing literature background.

I include all my comments, and I hope the authors are willing to take the time to make this work a solid publication that phage community can refer to.”

Response:

Thank you for appreciating our work. We have carefully considered all the suggestions and comments, and made revisions accordingly. All the changes made to the manuscript are marked up using the “Track Changes” in the word file of the revised manuscript. Please see our point-by-point response to the comments and concerns below:

Point 1:

“Please give the manufacturer of each reagent or medium used in this study?”

Response 1:

We thank you for this remark. We have added the missing information in the Materials and Methods section.

Thus, the following reagents were used to prepare lysogenic broth and LB agar: tryptone (Dia-M, Russia), yeast extract (Dia-M, Russia), agar-agar (PanReac Applichem, Germany).

In addition, in this study, we also used the following reagents:

  • PEG 8000 (PanReac Applichem, Germany);
  • NaCl (PanReac Applichem, Germany);
  • Imidazole (PanReac Applichem, Germany);
  • Glycerol (PanReac Applichem, Germany);
  • Tris-HCl (Sigma-Aldrich, USA)
  • Tris (Sigma-Aldrich, USA);
  • SDS (Serva, Germany);
  • glycine (Serva, Germany);
  • isopropyl β-D-1-thiogalactopyranoside (Sigma-Aldrich, USA)
  • EDTA (Sigma-Aldrich, USA);
  • Agarose Standard Low-mr (Bio-Rad, USA);
  • Gelatin (Difco, USA);
  • as well as other chemical compounds: NaH2PO4, Na2HPO4, ZnSO4, CsCl, MgCl2, MgSO4, CaCl2, NiCl2, CH3COOK, NaOH, HCl, chloroform (Reachim, Russia).

Point 2:

“Line 116: Enterococcus should be italicized.”

Response 2:

Thank you for pointing out typographical errors in our manuscript. We have corrected all the typos you mentioned.

Point 3:

“Line 123 – 125; 294-250: Why author used different buffers, with and without glycine? The presence/absence of glycine could influence phage pH stability. In my opinion, the authors should adjust with 1 M NaOH or HCl the phage cocktail to yield a pH range of 2–13, according to Lu and colleagues (2020)”

Response 3:

Indeed, in our first experiments on the pH stability of phages, we added a bacteriophage preparation (100 µl) to SM aliquots (900 µl) with pH adjusted with HCl or NaOH. However, we soon found that the addition of the bacteriophage preparations (which contained 50 mM Tris-HCl) brought pH back to neutral. Further, in all subsequent experiments, we decided to use a set of buffer solutions to stabilize pH (Kazantseva 2021, Kazantseva 2022). A set of buffer solutions are often used to determine the pH stability of viral particles, in particular T4 phage (Ward 1979), T2 phage (Hook 1946), T7 phage (Kirby 1949).

Regarding the effect of glycine. In the experiments presented in this paper, we used glycine-HCl buffer for pH 2.2 and 3 and glycine-NaOH buffer for pH 9 and 10. Figure 4b shows that at pH 9, in glycine-NaOH buffer, iF6 had the same stability as at pH 8 in sodium-phosphate buffer. Thus, it appears that glycine does not affect the stability of the iF6 phage.

  1. Kazantseva, O. A., Buzikov, R. M., Pilipchuk, T. A., Valentovich, L. N., Kazantsev, A. N., Kalamiyets, E. I., & Shadrin, A. M. (2021). The Bacteriophage Pf-10-A Component of the Biopesticide "Multiphage" Used to Control Agricultural Crop Diseases Caused by Pseudomonas syringae. Viruses, 14(1), 42. https://doi.org/10.3390/v14010042
  2. Kazantseva OA, Piligrimova EG, Shadrin AM. Novel Bacillus-Infecting Bacteriophage B13—The Founding Member of the Proposed New Genus Bunatrivirus. Viruses. 2022; 14(10):2300. https://doi.org/10.3390/v14102300
  3. Ward RL, Ashley CS. pH modification of the effects of detergents on the stability of enteric viruses. Appl Environ Microbiol. 1979 Aug;38(2):314-22. doi: 10.1128/aem.38.2.314-322.1979. PMID: 42351; PMCID: PMC243483.
  4. SHARP DG, HOOK AE, et al. Sedimentation characters and pH stability of the T2 bacteriophage of Escherichia coli. J Biol Chem. 1946 Sep;165(1):259-70. PMID: 21001207.
  5. KERBY GP, GOWDY RA, et al. Purification pH stability and sedimentation properties of the T7 bacteriophage of Escherichia coli. J Immunol. 1949 Sep;63(1):93-107. PMID: 18139410.

Point 4:

“Lines 262-266: This paragraph should be written in the results section.”

Response 4:

Thank you for your comment. We have taken into account your remark and the paragraph (Lines 262-266) was moved to the section 3. Results, the subsection 3.10. “Spectrum of lytic activity of endolysins”.

Point 5:

“Some analyses need a statistical analysis such as temperature, pH stability, between MOI and between endolysins. With no statistical analysis, we can not talk about differences only about the trend.”

Response 5:

Thank you for pointing this out. We have improved our manuscript by adding new subsection 2.17. “Statistical Analysis” in the 2. Materials and Methods section.

Point 6:

“Lines 288-289: 1 hour of incubation is insufficient to conclude how different temperatures can influence phage activity. We can talk only about stability for 1 hour. I suggest checking 1-week storage at 4, 20 and 37°C to determine stability in the human body temperature and storage conditions.”

Response 6:

In this study, we assessed the temperature range in which the phage and its enzymes are capable of exhibiting lytic activity. It is believed that if a phage or enzyme does not show activity after an hour of incubation at temperature less than 40°C (physiological body temperature), then they are of no practical interest, since, when using a drug based on a bacteriophage in therapy, a significant period is the action of the phage or its endolysins precisely in the first hour of treatment of the area affected by bacteria.

As you correctly noted, these data are not directly applicable to determine the optimal conditions and duration of storage. As for long-term storage, we did not have the task of determining the shelf life of the iF6 phage. However, based on our experimental data, we found that the purified phage preparation of iF6 phage can be stored at +4°C no more than for six months without a drop in phage titer. In addition, according to the manufacturer's instructions, the expiration date of Sextaphag® at a temperature of 2-8°C is 2 years.

Point 7:

“Lines 388: Please describe the content of genes responsible for lytic/lysogenic cycle – this is the most important criterium for phage selection for therapeutic purposes, as well as known genes encoding toxins, antibiotic resistance and virulence or superinfection immunity protein. The figure is insufficient to interpret by the reader.”

Response 7:

We cannot agree that figure 5 is insufficient for the reader to interpret. The iF6 genome map in Figure 5 shows all the genes responsible for the life cycle that have been identified.

The iF6 genome does not contain any genes encoding toxins, antibiotic resistance and virulence or superinfection immunity protein. More detailed information on the iF6 genome annotation is provided in Table S2. Annotation of Enterococcus phage iF6, file “Supplementary information”.

Point 8:

“Lines 442-455: Authors should add SDS-PAGE gels pictures to show aggregates and protein products of expression indicating their proper weight”

Response 8:

We have added SDS-PAGE gel with preparations of iF6 endolysins into Supplementary Information (Figure S4) and updated the Results section (“subsection: 3.8. Cloning and characterization of recombinant endolysins”).

Point 9:

“Line 485: Please add citations”

Response 9:

Thank you for the comment. We have improved the text in this subsection and added the appropriate references:

  • Uchiyama J., Rashel M., Maeda Y., Takemura I., Sugihara S., Akechi K., Muraoka A., Wakiguchi H., Matsuzaki S. Isolation and characterization of a novel Enterococcus faecalis bacteriophage phiEF24C as a therapeutic candidate. FEMS Microbiol Lett. 2008 Jan;278(2):200-6. doi: 10.1111/j.1574-6968.2007.00996.x. PMID: 18096017.
  • Khalifa L., Gelman D., Shlezinger M., Dessal A.L., Coppenhagen-Glazer S., Beyth N., Hazan R. Defeating Antibiotic- and Phage-Resistant Enterococcus faecalis Using a Phage Cocktail in Vitro and in a Clot Model. Front Microbiol. 2018 Feb 28;9:326. doi: 10.3389/fmicb.2018.00326. PMID: 29541067; PMCID: PMC5835721.
  • Kishimoto T., Ishida W., Nasukawa T., Ujihara T., Nakajima I., Suzuki T., Uchiyama J., Todokoro D., Daibata M., Fukushima A., Matsuzaki S., Fukuda K. In Vitro and In Vivo Evaluation of Three Newly Isolated Bacteriophage Candidates, phiEF7H, phiEF14H1, phiEF19G, for Treatment of Enterococcus faecalis Endophthalmitis. Microorganisms. 2021 Jan 20;9(2):212. doi: 10.3390/microorganisms9020212. PMID: 33498561; PMCID: PMC7909552
  • Tkachev, P.V.; Pchelin, I.M.; Azarov, D.V.; Gorshkov, A.N.; Shamova, O.V.; Dmitriev, A.V.; Goncharov, A.E. Two Novel Lytic Bacteriophages Infecting Enterococcus spp. Are Promising Candidates for Targeted Antibacterial Therapy. Viruses 2022, 14, 831. https://doi.org/10.3390/v14040831

Point 10:

“In the Results section authors describe endolysins HUgp82 and HUgp84, while in the Discussion section the names iF6_82 and iF6_84. It is not clear if we talk about the same proteins. Please use the same name everywhere.”

Response 10:

Thank you for your valuable comment. We have improved the text in the Materials and Methods section, subsection 2.12. “Endolysins genes cloning and expression, and endolysin purification” so that the readers have a perfectly clear understanding of the abbreviations that we used in the study (i.e. Gp82 and Gp84 for original proteins; Gp82-H and Gp84-H for proteins with C-terminal his-tag and HU-Gp82 and HU-Gp84 for ubiquitinylated forms). We have also added the amino acid sequences of the his-tag and ubiquitin domain.

For the genes encoding the endolysins we have specified locus tags identifiers (locus tag: iF6_82 and iF6_84).

Point 11:

“I suggest rewriting the discussion and starting with the overall characteristic of iF6 phage and then describing details concerning endolysins. It will build a more interesting story. Authors should discuss the great potential of HUgp84 in Enterococcus combating and its resistance to harsh conditions. Please compare it with other papers where authors indicate a higher ability of lysins than phages to be an antibacterial tool even if they are against other pathogens than Enterococcus.”

Response 11:

Thank you for your valuable suggestions which have helped us in improving our manuscript. Based on your advice, we have added all the necessary information to the Discussion section.

In addition, we updated all the figures corrected in accordance with the requirements of the Viruses journal:

Figure 1: The designations of panels A and B have been changed to (a) (b), a “cm” mark has been added next to the ruler.

Figure 2: The vertical axes has been adjusted for both panels. The vertical axes on panels (a) and (b) have been renamed to "Phage titer, PFU/ml". Both vertical axes have been set to the same scale.

Figure 3: The font size has been increased. The color scheme of the lines has been rearranged to be more readable.

Figure 4: The font size has been increased. For panel (b), the vertical axis has been renamed from “PFU/mL” to “Phage titer, PFU/mL”.

Figure 5: no changes.

Figure 6: All Latin names have been italicized.

Figure 7: Changed the red color legend in the color scheme from “DNA packaging and structural protein” to “DNA packaging and structural proteins”

Figure/ 8: The designations of panels A and B have been changed to (a) (b) and (c), text front changed to Palatino Linotype.

Figure 9: Arrow marks have been added to indicate when endolysins were added. The line label color scheme for lysis curves has been simplified. The font size has been increased.

Figure 10: The font size has been increased. The name of the horizontal axes has been changed from "pH" to "pH value" in panels (a) and (b).

Round 2

Reviewer 2 Report

Manuscript in its present form is can by published in Animals.